# DIFFERENTIATION OF MULTI-OBJECTIVE DATA-DRIVEN DECISION PIPELINE

## ABSTRACT

Real-world scenarios frequently involve multi-objective data-driven optimization problems, characterized by unknown problem coefficients and multiple conflicting objectives. Traditional two-stage methods independently apply a machine learning model to estimate problem coefficients, followed by solving the predicted optimization problem. The independent use of optimization solvers and prediction models may lead to suboptimal performance due to mismatches between their objectives. Recent efforts have focused on end-to-end training of predictive models that use decision loss derived from the downstream optimization problem. However, these methods have primarily focused on single-objective optimization problems, thus limiting their applicability. We aim to propose a multi-objective decision-focused approach to address this gap. In order to better align with the inherent properties of multi-objective optimization, we propose a set of novel loss functions. These loss functions are designed to capture the discrepancies between predicted and true decision problems, considering solution space, objective space, and decision quality, named landscape loss, Pareto set loss, and decision loss, respectively. Our experimental results demonstrate that our proposed method significantly outperforms traditional two-stage methods and most current decision-focused methods.

## 1 INTRODUCTION

Uncertain decision-making is prevalent in various real-life scenarios (Kotary et al., 2021). These scenarios involve a workflow for handling data-driven decision problems where parameter coefficients are predicted based on environmental or historical information, and decisions are made using these predictions. Most traditional approaches decompose this workflow into the prediction phase and the decision phase. Obtaining a perfect prediction model is often unachievable. The problem coefficients generated by the prediction model are frequently noisy. Given that conventional prediction models prioritize predictive accuracy, they often neglect the structure and attributes of downstream optimization problems. However, in most of optimization problems, the impact of problem coefficients on the final decision is not uniform. Without a decision-focused approach, decisions derived from imperfectly predicted coefficients are more prone to deviate from the optimal solution. Consequently, the separation of the prediction and decision phases often leads to suboptimal outcomes due to the misalignment of objectives between these phases.(Amos & Kolter, 2017). Addressing this issue, the integration of prediction models with decision problems for unified training has emerged as a promising direction across various domains (Kotary et al., 2022; Vlastelica et al., 2020).

The integration of optimization problems into deep learning frameworks poses significant challenges, particularly because the mapping from predicted coefficients to the optimal decision variable is non-differentiable. To address the aforementioned issue, numerous decision-focused methods have been proposed to train a predictive model by minimizing decision loss associated with the downstream optimization task (Kotary et al., 2021). Considering the diverse forms and complexity of real-world optimization problems, as well as the time-intensive aspect of iterative problem-solving, prior research has concentrated on two principal directions. The first direction seeks to broaden the applicability of the approach by formulating additional optimization problems of varied forms, employing differentiable operators (Amos & Kolter, 2017; Wilder et al., 2019). The second direction (Shah et al., 2022) entails the development of efficient decision surrogate functions to diminish training time or amplify decision-focused performance. However, previous studies

have predominantly concentrated on the domain of single-objective problems, with less emphasis on extending the learning paradigms into the domain of multi-objective optimization (MOP).

Numerous data-driven multi-objective problems exist in real-world scenarios (Gunantara, 2018). The MOP demonstrate increased complexity in the search space compared to their single-objective counterparts. Optimal solutions for multi-objective problems are frequently non-unique and correspond to a Pareto front. Additionally, the gradient of multiple objectives often exhibit conflicting directions. Given the challenges and complexity, we aim to propose a novel Multi-Objective Decision-Focused Learning (MoDFL) model to address these gaps.

To address the mentioned inquiries, we propose three decision-focused loss functions tailored for MOP. The proposed loss function involves three modules, which measure the distance of objective space, solution space as well as the decision quality of representative point. Specifically, we introduce a landscape loss based on the sample rank maximum mean discrepancy (sRMMD) to quantify the discrepancy in the objective space across optimization problems. The objective space is considered a manifold within high-dimensional space, represented by the set of objective vectors corresponding to the solutions. We propose a Pareto set loss to directly measure the distance within the Pareto set, aiming to circumvent the homogeneity that may impede model training in certain optimization problems. The decision loss is analogous to other decision-focused losses, utilizing the decision quality of a representative solution as the loss criterion. This representative solution is derived from employing the weighted sum method in this paper. Differentiation of the proposed loss function is achieved by reparameterization method and transforming the multi-objective problem into a corresponding single-objective one. Building on these modules, MoDFL integrates the predictive model and MOP into a unified pipeline, which enables end-to-end training of the system.

The remainder of this paper is structured as follows. Section II provides an overview of the related work. Section III details the formulation of the data-driven multi-objective problem and introduces pertinent definitions. Section IV provides a motivating example on multi-objective decision learning. Section V present three novel loss functions, explore the differentiation of MOP, and explain the integration of the aforementioned modules within MoDFL. The performance of MoDFL is compared to that of two-stage methods and other state-of-the-art DFL method in Section VI. Finally, we conclude this paper in Section VII.

## 2 RELATED WORK

### 2.1 DIFFERENTIATION OF OPTIMIZATION PROBLEMS

A plethora of studies has investigated the integration of prediction problems with downstream decision-making processes. The scope of this study includes methodologies such as Smart Predict-and-Optimize (SPO)(Elmachtoub & Grigas, 2022), End-to-End Optimization Learning(Kotary et al., 2021), and Decision-Focused Learning (DFL)(Mandi et al., 2021). The central aspect of this topic aims to the differentiable mapping from problem coefficients to optimal decision variables, a concept we denote as the decision gradient herein. Utilizing the decision gradient enables the formulation of many optimization problems as differentiable operators within gradient-based methods like neural networks. Much of the seminal literature in this field stems from the work of Amos (Amos & Kolter, 2017). This approach proposed in (Amos & Kolter, 2017) leveraged the Karush-Kuhn-Tucker (KKT) optimality conditions for quadratic programming and the implicit function theorem to construct decision gradients. Nevertheless, this method's limitation lies in its requisite for a full-rank Hessian matrix of the objective function, constraining its applicability. Subsequent research has addressed this constraint by employing elaborate techniques to generalize the methodology across a broader spectrum of optimization problems. For instance, Wilder et al. (2019) introduced one quadratic regularization term into the objective function to extend the method proposed by Amos (Amos & Kolter, 2017) to linear programming (LP). Ferber et al. (2020) extended DFL to mixed-integer programming (MIP), relaxing MIP to LP by applying a cutting plane at the root LP node and leveraging Wilder's method (Wilder et al., 2019) to enable end-to-end training. Xie et al. (2020) approached the top-K problem through the lens of the optimal transport problem, extending Mandi and Guns's work (Mandi et al., 2020) to tackle it.

Another category of methods has emerged from the smart predict-and-optimize (SPO) method (Elmachtoub & Grigas, 2022). This approach primarily focuses on optimization problems with linear

objectives and involves predicting the parameters of the optimization problem. It introduces a convex surrogate upper bound on the loss, facilitating an accessible subgradient method. Vlastelica et al. (2020) also explore linear objective problems, computing decision gradients by perturbing the predicted problem coefficients. Expanding on this, Niepert et al. (2021) improve the method by incorporating noise perturbations into perturbation-based implicit differentiation to maximize the posterior distribution. Notably, all these methods require iterative addressing optimization problems during training, incurring substantial computational costs. To mitigate this issue, Mulamba et al. (2020) implemented a solution cache to record solutions discovered during model training and devised a surrogate decision loss function based on contrastive loss to reduce the time spent solving optimization problems. Kong et al. (2022) employed an energy-based model to characterize decision loss, thereby reducing the overhead in end-to-end training by linking minimum energy with minimum decision loss. Shah et al. (2022) introduced the surrogate convex loss functions to alleviate the computational burden, such as WeightedMSE, etc. Mandi et al. (2021) leveraged the learning-to-rank concept to devise a proxy for decision loss and introduced four surrogate decision loss functions.

## 3 PROBLEM DESCRIPTION

This study concentrates on DFL within the context of MOP. The given data include the contextual information (features) $x_i$, the true coefficients $y_i$, as well as the formulation of a parametric MOP. The subscript $i$ denotes the index of each optimization problem instance. In the decision-focused setting, the coefficients $y_i$ are not known in advance but can be estimated with a machine learning model. The procedure is divided into two phases: prediction and optimization. The prediction phase involves estimating the problem coefficients based on $x_i$; The optimization phase need to solve the optimization problem where the coefficients are fixed as predicted result. The goal is to optimize the objective values of the solution obtained in the optimization phase, using the true problem coefficients.

During the prediction phase, the prediction model is represented as $m(\theta, x)$, and the predicted problem coefficients can be defined as $\hat{y}_i \equiv m(\theta, x_i)$. During the optimization phase, the MOP can be formulated as presented in Eq.1.

$$\min_{\pi, g(y, \pi) \leq 0} f(y, \pi) = [f_1(y^1, \pi), \cdots, f_T(y^T, \pi)] \tag{1}$$

Where $\pi$ denotes one feasible solution, $f(\cdot)$ denotes the $T$ objective functions, and $g(\cdot)$ denotes the constraint functions. With the predicted problem coefficients, its optimal solution is defined as $\hat{\pi}_i(\theta, x_i) \equiv \arg\min_{\pi, g(\hat{y}_i, \pi) \leq 0} f(\hat{y}_i, \pi)$. The problem under study can be formulated as follows:

$$\min_{\theta} L(x, y, \theta) = \mathbb{E}_{x_i, y_i \in \mathbb{D}}[f_1(y_i^1, \hat{\pi}_i), \cdots, f_T(y_i^T, \hat{\pi}_i)] \tag{2}$$

## 4 METHODOLOGY

### 4.1 DECISION SURROGATE LOSS

#### 4.1.1 LANDSCAPE LOSS

In single-objective optimization problems, the objective space is one-dimensional, where a partial ordering can characterize the relationship between the objective function values of two solutions. However, in MOP, the T-dimensional objective space renders the partial order relationship inadequate for representing the distance in objective space across different problems. Moreover, the stringent criteria for establishing Pareto dominance often result in numerous objective vectors lacking any dominance relationship, especially for many-objective optimization problems. To address these limitations, we analogize the multi-dimensional objective space to the manifold in high-dimensional space, such as images, video, or audio signals. We use the concept of neighborhood relations in high-dimensional spaces to clearly define how different sets of objectives are connected and to identify common patterns in their overall distribution. Specifically, we utilize the objectives of solutions found during training model to approximately represent the objective space, and employ the

sRMMD (Masud et al., 2023) to measure the distance between the objective spaces of different optimization problems. This proposed metric is referred to as the landscape loss function.

To introduce the concept of sRMMD, we initially discuss entropy-regularized optimal transport, which involves determining an optimal coupling $c$ between a source distribution $P$ and a target distribution $Q$. The $\prod(P,Q)$ represents the set of joint probability measures on the product space with marginal distributions $P$ and $Q$, where $\phi \in P$ and $\psi \in Q$. The entropy-regularized optimal transport problem is formulated in the Eq.3. The dual form of the previously described problem is provided in Eq.4, with its derivation detailed in (Genevay, 2019).

$$\min_{c \in \prod(P,Q)} \int \frac{1}{2}|\phi - \psi|^2 \, dc(\phi,\psi) + \varepsilon KL(c||P \otimes Q) \tag{3}$$

$$\max_{u,v} \int u(\phi)dP(\phi) + \int v(\psi)dQ(\psi) + \varepsilon -$$
$$\varepsilon \int \int exp\left[\frac{1}{\varepsilon}(u(\phi) + v(\psi) - \frac{1}{2}|\phi - \psi|^2)\right] dP(\phi)dQ(\psi) \tag{4}$$

where the maximization is over the pairs $u \in L_1(P), v \in L_1(Q)$. The optimal entropic potentials for $\varepsilon$ are the pair of functions $u_\varepsilon$ and $v_\varepsilon$ correspond to the functions that achieve the maximum in Eq.4. This dual formulation can be solved using methods such as gradient descent or Sinkhorn's algorithm (Cuturi, 2013). By executing a predetermined number of iterations and applying automatic differentiation in PyTorch or TensorFlow, we can compute the gradient of sRMMD. Building upon $u_\varepsilon$ and $v_\varepsilon$, we present the soft rank map in its sample-based form as follows:

$$R_\varepsilon^n(\phi) = \frac{\sum_{i=1}^N \psi_i exp(\frac{1}{\varepsilon}(v_\varepsilon^n(\psi_i) - \frac{1}{2}|\phi - \psi_i|^2))}{\sum_{i=1}^N exp(\frac{1}{\varepsilon}(v_\varepsilon^n(\psi_i) - \frac{1}{2}|\phi - \psi_i|^2))} \tag{5}$$

For a distribution $P$, the sample rank map $\mathbb{R}^m$ is defined as the plug-in estimate of the transport map from $P$ to a uniform distribution $Q = Unif([0,1]^d)$, where $d$ denotes the dimensionality. Let $k : \mathbb{R}^m \times \mathbb{R}^m \to \mathbb{R}$ be a characteristic kernel function. Let $R_{\tau,\varepsilon}(\phi)$ denote the soft rank map of $P_\tau$ for $\tau \in (0,1)$. The distribution $P_\tau$ is the mixture distribution of $P_X$ and $P_Y$, where $P_\tau = \tau P_X + (1-\tau)P_Y$. Samples $X_i, X_j$ are drawn from $P_X$; similarly, samples $Y_i, Y_j$ are drawn from $P_Y$. The sRMMD between the distribution of $P_X$ and $P_Y$ can is expressed as follows:

$$\text{sRMMD}_{\tau,\varepsilon}^{m,n}(P_X,P_Y)^2 \cong \frac{1}{m^2}\sum_{i,j=1}^m k(R_{\tau,\varepsilon}^{m+n}(X_i), R_{\tau,\varepsilon}^{m+n}(X_j))$$
$$+ \frac{1}{n^2}\sum_{i,j=1}^m k(R_{\tau,\varepsilon}^{m+n}(Y_i), R_{\tau,\varepsilon}^{m+n}(Y_j)) -$$
$$\frac{2}{nm}\sum_{i=1}^m\sum_{j=1}^n k(R_{\tau,\varepsilon}^{m+n}(X_i), R_{\tau,\varepsilon}^{m+n}(Y_j)) \tag{6}$$

Assume we have a pooling of solutions $S$, where $\pi_i \in S$. The objective vectors for the predicted and true problems is represented as $\phi_i, \psi_i$, respectively, where $\phi_i, \psi_i \in \mathbb{R}^{|S| \times T}$ and $i$ denotes the index of the optimization problem instance

$$L_l(x,y,\theta) = \frac{1}{N}\sum_i \text{sRMMD}_{\tau,\varepsilon}^{|S|,|S|}(f(y,\pi_i), f(m(\theta,x),\pi_i)) \tag{7}$$

### 4.1.2 PARETO SET LOSS

Optimal solutions may differ across optimization problems. However, certain optimization problems, despite having different coefficients, can exhibit identical landscapes and share the same optimal solution. For example, a normalization for the coefficients of LP can preserve both the optimization landscape and solution optimality. In light of this homogeneity, we introduce a loss function that directly measures the distance between solution spaces of optimization problems. With a focus on optimization problems that target optimal solutions, our proposed loss function quantifies the disparity between sets of optimal solutions. Drawing on the concept of inverted generational distance, this loss function employs the minimum distance between the Pareto sets of the predicted and true problems in solution space. Let $PS_y^*$ represent the Pareto set of the true problem and $PS_{\hat{y}}^*$ denote the Pareto set of the predicted problem. The proposed loss function is approximated by $min_{\pi_y^* \in PS_y^*} d(\hat{\pi}, \pi_y^*)$, which is represented as follow:

$$L_{ps}(x, y, \theta) = \frac{1}{N} \sum_i min_{\pi_{y_i}^* \in PS_{y_i}^*} d(\hat{\pi}, \pi_{y_i}^*) \tag{8}$$

### 4.1.3 DECISION LOSS

We also adopt the decision quality associated with the Pareto optimal solutions of the predicted problem as the loss function. However, due to the multidimensional nature of the Pareto front, the aforementioned loss function cannot be directly applied in the multi-objective scenario. Here, we transform the predicted problem and the true problem into single-objective problems. Subsequently, we employ the optimal solutions of the transformed predicted problem in the objective function of the transformed true problem as the loss function, which is named as decision loss function.

We apply the conventional weighted sum method to transform multi-objective problem into single-objective problem. Due to the difference in the magnitude of different objectives, the predicted coefficients are initially processed through an instance normalization layer. The process is detailed as follows:

$$BN(\hat{y_i^j}) = \frac{\hat{y_i^j} - mean(\hat{y_i^j})}{std(\hat{y_i^j})}; j = 1, \cdots, T \tag{9}$$

where $mean(\hat{y_i^t})$ and $std(\hat{y_i^t})$ denote the mean and standard deviation of $\hat{y_i^t}$ respectively. In optimization problems with LP and MIP, it's easy to prove that the instance normalization layer preserves the relative cost value ordering. Consequently, the normalization layer maintains the optimization landscape and the optimality of solutions unchanged.

Equipped with uniform weight, we focused the weighted optimization problem $f_w(\hat{y}, \pi) = \frac{\sum_j f^j(BN(\hat{y^j}), \pi)}{T}$. This method ensures that the mapping from problem coefficients to the optimal solution aligns with that of a single-objective decision-focused method. Let $\hat{\pi}$ denote as the optimal solution of $f_w(\hat{y}, \pi)$. The decision loss is then articulated as follows:

$$L_d(x, y, \theta) = \frac{\sum_i \sum_j f(BN(y_i^j), \hat{\pi}_i)}{NT} \tag{10}$$

## 4.2 DIFFERENTIATION OF OPTIMIZATION MAPPINGS

Differentiation of optimization mappings refers to the process of computing the gradient of the decision surrogate loss with respect to the predicted problem coefficients. As previously discussed, we can observe that the decision surrogate loss is a function of $y$, $\hat{y}$, and $\hat{\pi}$. When $\hat{\pi}$ is sampled from the given distribution of solution (or solution cache), the gradient of the decision surrogate loss on the search space loss function can be calculated using reparameterization techniques (Kingma et al., 2015). Accordingly, the gradient of the landscape loss function is determined in this manner. For the gradients of Pareto set loss function and decision loss function, the decision gradient of optimization mappings is decomposed into two terms by the chain rule, as expressed below:

$$\frac{\partial L(x, y, \theta)}{\partial \hat{y}} = \frac{\partial L(x, y, \theta)}{\partial \hat{\pi}(\theta, x)} \frac{\partial \hat{\pi}(\theta, x)}{\partial \hat{y}} \tag{11}$$

The first term denotes the gradients of the decision loss with respect to the decision variable. In this work, the proposed decision loss function are both the continuous function on the decision variable, facilitating automatic differentiation by deep learning frameworks. The second term corresponds to the gradients of the optimal decision with respect to the predicted coefficient. This term involves the non-differentiable $argmin$ operator. To overcome this issue, various efficient surrogate functions and carefully designed techniques have been proposed. With the prevalent use of linear programming in practical applications, we demonstrate our approach through data-driven linear programming. The differentiation of the optimal condition for smooth linear programming (DSLP) (Wilder et al., 2019) is utilized to derive the gradient of the optimal decision relative to the problem coefficients. The derivation process can be referred to in the appendix.

### 4.3 APPROACH: MULTI-OBJECTIVE DECISION FOCUSED LEARNING

During the training phase of MoDFL, the input dataset comprises relevant feature $x_i$, true problem coefficient $y_i$, gradients of the cost function with respect to solutions $\nabla_\pi f(y_i, \pi)$ as well as a set of Pareto optimal solutions $P_{y_i}^*$. As the modules within the proposed method are differentiable,

---

**Algorithm 1** Multi-Objective Decision Focused Learning

**Input:** $x$; $y$; $P_{y_i}^*$;
**Output:** Prediction Model: $m(\theta, \cdot)$;
1: **for** epoch k =0, 1,... **do**
2:    **for** instance i =0, 1,... **do**
3:       $\hat{y}_i \leftarrow m(\theta, x_i)$
4:       employ instance normalization and weight-sum method to generate $f_w(\hat{y}, \pi)$
5:       employ DSLP to generate the $\hat{\pi}_i$
6:       **if** random() $\leq p_{solve}$ **then**
7:          Obtain solutions $\pi_i^{\hat{new}}$ by invoking a multi-objective solver for Eq.10
8:          $S \leftarrow S \cup \{\pi_i^{\hat{new}}\}$
9:       **end if**
10:      Calculate $L_{all}(x, y, \theta)$ according to Eq. 12
11:      Update model parameter $\theta$ according to back propagation algorithm
12:    **end for**
13: **end for**
14: **return** Prediction Model: $m(\theta, \cdot)$;

---

we focus on the forward pass of the neural network. Initially, prediction model $m(\theta, \cdot)$, produces multiple group of problem coefficients $\hat{y}_i$ according to the $x_i$. The mentioned prediction model may refer to one multi-task model or multiple single-task models. Secondly, we transform the studied MOP problem to a single-objective problem, and employ DSLP method to generate the differentiable solution $\hat{\pi}_i$. Thirdly, we update the solution cache via adding the optimal solution of predicted problem with a certain probability. Finally, we calculate the decision surrogate loss according to $y$, $\hat{y}$, and $\hat{\pi}$, where the final loss function is weighted sum of the overall loss function on objective space, solution space as well as the decision quality of representative point.

$$L_{all}(x, y, \theta) = \lambda_l L_l(x, y, \theta) + \lambda_d L_d(x, y, \theta) + \lambda_{ps} L_{ps}(x, y, \theta) \quad (12)$$

where $\lambda_l$, $\lambda_d$, $\lambda_{ps}$ denotes the hyper-parameters. The solution cache $S$ is utilized to calculate the landscape loss; The solution $\hat{\pi}_i$ generated by DSLP is utilized to calculate the Pareto set loss and decision loss. Given that all modules are differentiable, the implementation of the backward pass is straightforward. The detail pseudo code is presented in Algorithm 1.

## 5 EXPERIMENT

### 5.1 BENCHMARK PROBLEM

We firstly explore a particular case of web advertisement allocation within the Anonymous App, aiming to optimize overall click metrics and enhance user visitation on the following day. Unlike

traditional recommendation systems, our approach allocates a single advertisement per user query, each ad belonging to a unique business category. Our objective is to regulate the ad display frequency from each category to align with a predetermined parameter over a set period. This scenario is treated as an online matching optimization problem, typically resolved using a primal-dual method. The problem's formalization involves predicting click-through and re-login probabilities, for subsequent user engagement. The optimization seeks to balance exposure across business categories within specified thresholds. For experimental validation, we selected a subset of 30,000 queries to generate instances for testing our MOP, highlighting the problem's complexity in a decision-focused context.

Besides, we address a MOP problem inspired by a benchmark issue, utilizing data from the Cora dataset, which comprises 2708 scientific papers (nodes) interconnected by citations (edges). The challenge involves partitioning the network into 27 sub-graphs, each containing 100 nodes, using the METIS algorithm. This setup facilitates the formation of bipartite graphs for each instance, aiming to maximize inter-set citation links. Our objective diverges towards generating alternative objective values, representing perturbed citation relationships, to evaluate the model's adaptability to slight deviations in data. This approach transforms the problem into a LP problem suited for bipartite matching, focusing on optimizing the proposed objectives while adhering to set constraints. Detailed explanations of our benchmark Problem are documented in the appendix.

## 5.2 EXPERIMENTAL SETUP AND BASELINE METHOD

### 5.2.1 BASELINE METHOD

The baseline methods under consideration include a straightforward two-stage approach and several state-of-the-art decision-focused approaches. Given that current DFL methods are single-objective, we implement the baseline methods by substituting their loss functions with uniformly weighted loss functions, i.e., $L(x, y, \theta) = \sum_j \frac{L(x, y^j, \theta)}{T}$. The $L(x, y^j, \theta)$ represents the loss function of $j_{th}$ objective in studied decision problem. The prediction model and solver are identical with to those in our proposed method. For decision-focused methods, the gradient of decision variable with regard to problem coefficient $\frac{\partial \hat{\pi}}{\partial y^j}, j = 1, \cdots, T$ is the same as single-objective methods. The key difference lies in the decision loss function and the technique used to compute the gradient of the problem coefficients with respect to the decision loss function. Specifically, the baseline method includes the following:

- **TwoStage**:employs a prediction model with an independent solver as the two-stage baseline.

- **SPO**(smart predict and optimize) (Elmachtoub & Grigas, 2022): utilises the surrogate loss function proposed by Elmachtoub et al.

- **BB** (Vlastelica et al., 2020): calculates the decision gradient by differentiation of blackbox combinatorial solvers.

- **MAP**(Niepert et al., 2021): employs the surrogate loss function which incorporate noise perturbations into perturbation-based implicit differentiation and maximizing the resulting posterior distribution .

- **NCE** (Mulamba et al., 2020): uses solution caching and contrastive losses to construct surrogate loss function.

- **Pointwise/Listwise** (Mandi et al., 2021): employs the surrogate loss function derived from the technique of learning to rank.

Our study prioritizes decision quality within the paradigm of DFL. Accordingly, the $p_{solve}$ parameter, the probability of invoking optimization solver, is uniformly set to 1 for the MAP, NCE, Pointwise, and Listwise methods[1], acknowledging that methods with a $p_{solve}$ closer to 1 are typically associated with enhanced performance.

---

[1]https://github.com/JayMan91/ltr-predopt

Table 1: Experimental results on the dataset of Bipartite matching among scientific papers.

| Method | GD | MPFE | HAR | $r_1$ | $r_2$ | $r$ |
|---|---|---|---|---|---|---|
| BB | 12.6335 | 40.0616 | 1.0830 | 0.9317 | 0.5616 | 0.7466 |
| MAP | 15.6359 | 43.9498 | 1.1736 | 0.9488 | 0.7181 | 0.8335 |
| NCE | 12.5653 | 40.1353 | 1.0791 | 0.9355 | 0.5579 | 0.7467 |
| Listwise | 12.0901 | 39.4815 | 1.0848 | 0.9342 | 0.5355 | 0.7348 |
| Pointwise | 12.1347 | 39.8968 | 1.0872 | 0.9320 | 0.5413 | 0.7366 |
| SPO | 12.5224 | 40.0768 | 1.0840 | 0.9376 | 0.5563 | 0.7470 |
| Twostage | 12.2893 | 39.4263 | 1.0910 | 0.9309 | 0.5443 | 0.7376 |
| MoDFL | **11.8545** | **39.0535** | **1.0707** | **0.9263** | **0.5261** | **0.7262** |

Table 2: Experimental results on the dataset of Bipartite matching among scientific papers.

| Method | GD | MPFE | HAR | $r_1$ | $r_2$ | $r$ |
|---|---|---|---|---|---|---|
| BB | 12.6335 | 40.0616 | 1.0830 | 0.9317 | 0.5616 | 0.7466 |
| MAP | 15.6359 | 43.9498 | 1.1736 | 0.9488 | 0.7181 | 0.8335 |
| NCE | 12.5653 | 40.1353 | 1.0791 | 0.9355 | 0.5579 | 0.7467 |
| Listwise | 12.0901 | 39.4815 | 1.0848 | 0.9342 | 0.5355 | 0.7348 |
| Pointwise | 12.1347 | 39.8968 | 1.0872 | 0.9320 | 0.5413 | 0.7366 |
| SPO | 12.5224 | 40.0768 | 1.0840 | 0.9376 | 0.5563 | 0.7470 |
| Twostage | 12.2893 | 39.4263 | 1.0910 | 0.9309 | 0.5443 | 0.7376 |
| MoDFL | **11.8545** | **39.0535** | **1.0707** | **0.9263** | **0.5261** | **0.7262** |

### 5.2.2 EVALUATION METRIC

We evaluate the performance of algorithms by the quality of their decisions in relation to a true optimization problem. While the objective function value is a direct measure, it may be skewed by differing scales across objectives. Therefore, we adopt the average percentage regret as a key metric to judge performance, with a lower value indicating superior outcomes. Additionally, we consider three other metrics commonly employed in MOP analyses: generational distance (GD) (Ishibuchi et al., 2015), maximum Pareto front error (MPFE), and hyper area ratio (HAR). GD gauges the proximity of the predicted Pareto front to the true front, MPFE measures the maximal deviation of approximate solutions from the optimal set, and HAR assesses the relative coverage area of the Pareto fronts. Collectively, these metrics help determine the efficacy of the proposed solutions in approximating the true Pareto front. Detailed formulations and computational approaches for these metrics are provided in the Appendix.

### 5.2.3 EXPERIMENTAL SETUP ON PREDICTION MODEL AND SOLVER

The prediction model employed in the Web Advertisement Allocation experiment is the multi-gate mixture-of-experts (MMOE), a typical model in the field of computational advertising (Ma et al., 2018). In the bipartite matching experiment with scientific papers, we utilized four-layer fully connected neural networks to predict the multiple groups of problem coefficients. The configuration and architecture of the neural network followed the specifications in (Wilder et al., 2019). As for the tested optimization problem, we adopted the weighted-sum method to convert the MOP problem into multiple single-objective problems and utilized the HiGHS solver to address the transformed single-objective problem. Further details on the methodological aspects and normalization process are elaborated upon in the appendix.

### 5.2.4 CONFIGURATION ON DECISION-FOCUSED SETTING

The learning rate was configured to $10^{-1}$ following the setting in Wilder et al. (2019). The batch size was set to 8. The hyper-parameters in Eq.12 were specified as $\lambda_l = 1; \lambda_d = 2; \lambda_{ps} = 5$. Early stopping was implemented, terminating the training loop if the validation set loss did not improve for 5 consecutive epochs. The maximum number of epochs was set to 50. The initial squared regularization term $\gamma$ was set to 0.35. All experimental comparisons were carried out on an A100 cluster

Table 3: Experimental results on data-driven decision problems with three objectives.

| Method | GD | MPFE | HAR | $r_1$ | $r_2$ | $r_3$ | $r$ |
|--------|------|------|------|------|------|------|------|
| BB | 9.6199 | 47.3243 | 1.2211 | 0.9300 | 0.5395 | 0.7395 | 0.7363 |
| MAP | 11.1703 | 48.6628 | 1.3321 | 0.9484 | 0.6329 | 0.7867 | 0.7894 |
| NCE | 9.6434 | 47.5465 | 1.2226 | 0.9311 | 0.5427 | 0.7406 | 0.7381 |
| Listwise | 9.6947 | **46.9460** | 1.2325 | 0.9252 | 0.5445 | 0.7358 | 0.7352 |
| Pointwise | 9.7053 | 47.4230 | 1.2352 | 0.9353 | 0.5409 | 0.7400 | 0.7387 |
| SPO | 10.2388 | 50.2437 | **1.2003** | 0.9298 | 0.5801 | 0.7474 | 0.7524 |
| TwoStage | 9.8151 | 48.4521 | 1.2103 | 0.9257 | 0.5520 | 0.7402 | 0.7393 |
| MoDFL | **9.5605** | 47.1387 | 1.2088 | **0.9243** | **0.5367** | **0.7285** | **0.7298** |

Table 4: The performance of different landscape loss function.

| Method | GD | MPFE | HAR | $r_1$ | $r_2$ | $r$ |
|--------|------|------|------|------|------|------|
| MMD | 11.9022 | 39.4382 | 1.0793 | 0.9365 | 0.5275 | 0.7320 |
| DSPM | 12.4058 | 39.4131 | 1.0840 | 0.9307 | 0.5504 | 0.7405 |
| sRMMD(Ours) | **11.8545** | **39.0535** | **1.0707** | **0.9263** | **0.5261** | **0.7262** |

environment and repeated 5 times for consistency. We employ a 6-dimensional Gaussian mixture kernel (Masud et al., 2023) as our kernel function in landscape loss, with bandwidth parameters $\sigma = (1, 2, 4, 8, 16, 32)$. The associated hyperparameters, $\tau = 0.5, \varepsilon = 10^{-5}$.

## 5.3 VALIDATING THE PERFORMANCE OF DFL IN MULTI-OBJECTIVE PROBLEM

### 5.3.1 WEB ADVERTISEMENT ALLOCATION

In the Table I, we compare the performance of different methods using various evaluation metrics. The MoDFL achieves a GD value of 0.6416, which is the lowest among all methods. It also outperforms other methods in terms of MPFE, HAR, $r$ and $r_1$, with competitive results in $r_2$. Among the competing methods, Listwise shows the second-best performance with the best $r_2$ value. These results indicate that MoDFL is superior in terms of minimizing GD and other metrics. Overall, the experimental comparison demonstrates the efficacy of MoDFL in achieving better performance on multiple evaluation metrics, highlighting its potential in addressing the data-driven multi-objective problem. Compared to single-objective problems, MoDFL still presents significant challenges. Applying a weighted average of the decision losses in single-objective DFL may deteriorate algorithm performance. As evident from the results of $r_1$ and $r_2$, MAP, Pointwise, and BB fail to effectively balance multiple objectives, resulting in a significant degradation of performance in some objectives. In order to capitalize on the benefits afforded by end-to-end training, it is imperative to propose appropriate methodologies for addressing multi-objective issues within the paradigm of DFL.

### 5.3.2 BIPARTITE MATCHING AMONG SCIENTIFIC PAPERS

Based on the experimental comparison data presented above, we evaluated several methods including BB, MAP, NCE, Listwise, Pointwise, SPO, twostage, and our proposed MoDFL on the dataset of bipartite matching among scientific papers. Among the methods tested, MoDFL consistently outperformed the others baseline in terms of all evaluation metrics. MoDFL achieved the lowest values for GD, MPFE , HAR, $r_1$ , $r_2$ , and $r$, indicating its superior performance in terms of decision quality. In comparison, the BB, MAP, NCE, Listwise, Pointwise, SPO, and Twostage methods exhibited slightly higher values across all evaluation metrics, indicating their inferior performance when compared to MoDFL. Overall, these results demonstrate that our proposed method MoDFL outperforms the existing methods in this study, highlighting its potential for improving various aspects of MOP. It is worth noting that in this set of experiments, the values for $r_1$, $r_2$, and $r$ are much higher compared to the previous experiments, indicating that the difficulty of the test problems in this case is relatively higher (Wilder et al., 2019). This also demonstrates that our proposed method, MoDFL, can effectively handle data-driven real-world problems of varying difficulty levels.

Table 5: Ablations on MoDFL.

| Method | GD | MPFE | HAR | $r_1$ | $r_2$ | $r$ |
|---|---|---|---|---|---|---|
| w/o Decision Loss | 12.5077 | 39.8507 | 1.0878 | 0.9300 | 0.5547 | 0.7424 |
| w/o Lanscape Loss | 12.0893 | 39.1558 | 1.0833 | 0.9314 | 0.5352 | 0.7333 |
| w/o Pareto Set Loss | 12.2063 | 39.5433 | 1.0765 | 0.9329 | 0.5427 | 0.7378 |
| MoDFL | **11.8545** | **39.0535** | **1.0707** | **0.9263** | **0.5261** | **0.7262** |

### 5.3.3 THE IMPACT OF NUMBER OF OBJECTIVES

The decision problems in tested experiments are bi-objective problem. In order to investigate the impact of number of objectives, we add one objective into the second benchmark and compare the performance of two-stage method and MoDFL. The third objective is the weighted sum of the first two objectives, where the weights used in this case are drawn from $U[0, 1]$. The experiment results are exhibited as Table. III. Similar to the results on the bi-objective problems, the testing results demonstrate that MoDFL achieves competitive performance. MoDFL outperforms all other algorithms in terms of GD, $r_1$, $r_2$, and $r_3$, $r$ while only slightly lagging behind Listwise in terms of MPFE. The HAR yielded by MoDFL is slightly lower than SPO. These findings demonstrate that the proposed MoDFL method is capable of effectively handling multi-objective problems.

### 5.3.4 THE CHOICE OF LANDSCAPE LOSS

To further elucidate our choice of sRMMD as the landscape loss function in MoDFL, we conducted experiments to compare the effects of replacing sRMMD with differentiable Spearman correlation coefficients(DSPM) Blondel et al. (2020) and the maximum mean discrepancy (MMD) (Gretton et al., 2012) in MoDFL. This experiment is conducted in the second benchmark. The comparison results presented in the Table V demonstrates the performance of different methods. In contrast, the DSPM method demonstrates marginally inferior performance across these metrics. However, the sRMMD approach, integrated within the MoDFL, surpasses the alternative strategies across all performance indicators, including GD, MPFE, HAR, and the metrics on decision regret. The comparative results underscore the enhanced performance of the MoDFL model when utilizing sRMMD as the landscape loss function, relative to the other methods assessed. It is evident that the adoption of sRMMD within the MoDFL framework contributes to a marked improvement in the model's performance as reflected by the evaluation metrics.

### 5.3.5 ABLATION STUDIES

To validate the individual components of our proposed MoDFL method, we conducted ablation experiments by removing those proposed components in the second benchmark problem. Specifically, we tested the effects of removing the decision Loss, landscape loss, and Pareto set loss from MoDFL. The comparative results are shown in Table VI. MoDFL achieved the best results across all metrics, indicating the importance of each loss function in MoDFL. Removing any of these components weakened the algorithm's performance. Among the three loss function, the decision Loss had the most significant impact on the overall performance. The effects of Pareto set loss were relatively is close to that of landscape loss. This suggests that for DFL in MOP, the surrogate loss function needs to align with the properties of multiple objectives and accurately measure the distance between the prediction problem and the true problem in the solution and search spaces.

## 6 CONCLUSION

Multi-objective data-driven problems are prevalent in real world. we consider one case which problem coefficients are unknown in advance and need to be estimated with machine learning models. We proposed one novel multi-objective decision-focused model to considering the prediction problem with the downstream MOP problem. Specifically, we propose one set of decision-focused loss function for MOP problem. The proposed loss function involves three parts, the decision loss, landscape loss, and Pareto set loss, which measure the distance of objective space, solution space as well as the decision quality of representative solution. Finally, experimental results show that our

proposed method has significant superiority over two-stage methods and the state-of-art methods. Current research in MOP and DFL remains limited. We plan to investigate more effective multi-objective decision-focused methods, and apply MoDFL to more forms of data-driven problems.

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
