# DIFFERENTIATION OF MULTI-OBJECTIVE DATA-DRIVEN DECISION PIPELINE

## A  APPENDIX

### A.1  THE DERIVATION ON DIFFERENTIATION OF OPTIMIZATION MAPPINGS

We illustrate our methodology using data-driven linear programming as an example. Notably, our approach extends to various problem types, provided that a suitable differentiable operator (Mandi et al., 2023) is substituted for DSLP. For a clearer exposition of DSLP, we first delineate linear programming with the subsequent equations:

$$\min_{\pi} \quad f(y, \pi) = [y^1\pi, \cdots, y^T\pi]$$
$$\text{s.t.} \quad A\pi \leq b \tag{1}$$

By leveraging the KKT conditions, the DSLP constructs a system of linear equations based on the predicted coefficient and the optimal decision. Applying the implicit function theorem allows us to derive the expression for the second term.

$$\begin{bmatrix} \nabla^2_{\hat{\pi}} f(y, \hat{\pi}) & A^T \\ diag(\lambda)A & diag(A\hat{\pi} - b) \end{bmatrix} \begin{bmatrix} \frac{\partial \hat{\pi}}{\partial \hat{y}} \\ \frac{\partial \hat{\lambda}}{\partial \hat{y}} \end{bmatrix} = \begin{bmatrix} \frac{\partial \nabla_{\hat{\pi}} f(y, \hat{\pi})}{\partial y} \\ 0 \end{bmatrix} \tag{2}$$

where $\lambda$ denotes the optimal dual variable of studied problems and $diag(\cdot)$ creates a diagonal matrix from an input vector. Due to the Hessian matrix of linear programming isn't full-rank, it fails to apply in the domain of linear programming. Wilder (Wilder et al., 2019) proposed to add one small squared regularizer term into LP which addressed the ill-conditioned Hessian matrix of LP. The objective function of Eq.1 in training phase is replaced with $f(y, \pi) = [y^1\pi + \gamma\|\pi\|_2^2, \cdots, y^T\pi + \gamma\|\pi\|_2^2]$. The second term can be calculated by solving the following system of linear equations.

$$\begin{bmatrix} 2\gamma I & A^T \\ diag(\lambda)A & diag(A\hat{\pi} - b) \end{bmatrix} \begin{bmatrix} \frac{\partial \hat{\pi}}{\partial \hat{y}} \\ \frac{\partial \hat{\lambda}}{\partial \hat{y}} \end{bmatrix} = \begin{bmatrix} I \\ 0 \end{bmatrix} \tag{3}$$

In the above equations, $\hat{\pi}, \lambda$ correspond to the optimal primal variable and dual variable and can be calculated by solving the quadratic programming problem derived from the linear programming. During the inference phase, the regularization factor $\gamma$ is set as 0 to yield an integral decision.

### A.2  THE MATHEMATICAL FORMULATIONS OF BENCHMARK PROBLEMS

#### A.2.1  WEB ADVERTISEMENT ALLOCATION

We examine one specific case of web advertisement allocation existing in anonymous App. The system is designed to optimize cumulative click metrics and increment the subsequent day's user visitation. As for each query, we recommend at most one advertisement to user. For each user query, a singular advertisement recommendation is proposed. This framework diverges from traditional recommendation systems in that each advertisement is associated with a distinct business category. Over a specified time frame, the display frequency of advertisements from any given business category is intended to approximate a pre-determined parameter $\delta$. The decision problem can be regard as one online-matching optimization problem, commonly addressed using the primal-dual approach. The formulation of this problem can be formulated as Eq.4. Within this equation, the cost vector $y^1, y^2$ denote the predicted the click-through probabilities and re-login probabilities for users

on the subsequent day, respectively. Let $i$ represent the query index, $j$ the candidate advertisement index, and $k$ the business category index, with $c(j)$ indicating the business category of item $j$. The target of exposure ratio for advertisements is specified by the vector $\delta$, with an allowable deviation encapsulated within the threshold $thr$, $thr > 0$. Furthermore, $ND, NC$ denote the quantities of queries and candidate advertisements, respectively.

$$
\begin{aligned}
\max_{\pi} \quad & f(y, \pi) = [\sum_{i,j} y_{ij}^1 \pi_{ij}, \sum_{i,j} y_{ij}^2 \pi_{ij}] \\
\text{s.t.} \quad & \sum_j \pi_{ij} \le 1; i = 1, 2, \cdots, ND \\
& \frac{\sum_{j,c(j)=k} \sum_i \pi_{ij}}{ND} \le \delta_k + thr; k = 1, 2, \cdots, NC \\
& \frac{-\sum_{j,c(j)=k} \sum_i \pi_{ij}}{ND} \le -\delta_k + thr; k = 1, 2, \cdots, NC \\
& \pi_{ij} \in \{0, 1\}
\end{aligned}
\tag{4}
$$

It is important to recognize that counter-factual outcomes are unattainable. A model was trained utilizing a dataset exceeding 20 million queries, employing the predictive output as labels. During the experimental phase, a random subset of 30,000 queries was selected to create 300 instances, with each instance consisting of 100 queries and 53 candidate advertisements. In the decision-focused setting, the prediction problem is one typical multi-task binary classification problem, incorporating a click-through rate prediction task, a prevalent and extensively researched problem within the domain of recommendation systems.

### A.2.2 BIPARTITE MATCHING AMONG SCIENTIFIC PAPERS

We adapted the benchmark problem proposed in (Wilder et al., 2019) to create multi-objective benchmark problem. The data were obtained from the cora dataset (Sen et al., 2008). In the dataset, each node corresponds to a scientific paper, and each edge represents a citation. The feature vector of nodes indicate the presence or absence of each word from a defined vocabulary. The dataset includes 2708 nodes. Wilder et al. employed the METIS algorithm (Karypis & Kumar, 1998) to partition the complete graph into 27 sub-graph, each with 100 nodes. Each graph corresponds to one instance. Subsequently, nodes within each instance were allocated to two sets of a bipartite graph, each comprising 50 nodes, to maximize the number of edges between the sets. More detail can refer to (Wilder et al., 2019).

The core is to generate the labels of an alternative objective value that differ from but are similar to the original labels. The cited relationship is denoted by $y^1$. We perturb the $y^1$ to generate $y^2$.

$$
y_{ij}^2 = I(r_{ij} \ge \rho)(1 - y_{ij}^1) + I(r_{ij} < \rho)y_{ij}^1
\tag{5}
$$

where $r_{ij}$ is draw from the uniform distribution from 0 to 1, $I(\cdot)$ is the indicator function, and $\rho$ is given constant to control the similarity between $y^1$ and $y^2$. In this study, $\rho$ is set as 0.05. The constrain function aligns with the conventional bipartite matching problem. Considering that the matching weight are positive and decision variables belong to $0, 1$, we can relax the problem to a linear programming formulation, as presented in Eq. 1. The $NU, NV$ represent the number of nodes in the left and right subsets of the bipartite graph, respectively.

$$
\begin{aligned}
\max_{\pi} \quad & f(y, \pi) = [\sum_{i,j} y_{ij}^1 \pi_{ij}, \sum_{i,j} y_{ij}^2 \pi_{ij}] \\
\text{s.t.} \quad & \sum_j \pi_{ij} \le 1; i = 1, 2, \cdots, NU \\
& \sum_i \pi_{ij} \le 1; j = 1, 2, \cdots, NV \\
& \pi_{ij} \in [0, 1]
\end{aligned}
\tag{6}
$$

## A.3 THE CALCULATION OF EVALUATION METRIC

In the setting of DFL, the output involves a set of solutions. We assess the algorithm's performance by measuring its decision quality against a true optimization problem. A straightforward metric is the objective function value of the true problem. However, owing to the variance in scale among objectives, we utilize the average percentage regret $r$ as the evaluation metric. The method of calculation is as follows:

$$r_j = \frac{1}{N} \sum_i \frac{f_j(y^j, \pi_i) - f_j(y^j, \pi^{*,j})}{f_j(y^j, \pi^{*,j})} \tag{7}$$

$$r = \frac{1}{T} \sum_j r_j \tag{8}$$

where $N$ denotes the size of solution set, and $\pi^{*,j}$ represents the optimal solution for the $j_{th}$ objective. A lower average percentage regret indicates better performance.

Besides, we consider three performance metrics widely used in the field of MOP problem. We denote the Pareto front of predicted and true problem, desperately $\hat{PS}$ and $PS^*$. The generational distance (GD) (Ishibuchi et al., 2015) measures the minimum distance between the Pareto front of predicted and true problem. The GD is defined as follows:

$$GD(\hat{PS}, PS^*) = \frac{\sum_{p \in \hat{PS}} d(p, PS^*)}{|\hat{PS}|} \tag{9}$$

The $d(p, PS^*)$ denotes the minimum Euclidean distance between $p$ and the points in $PS^*$. A lower GD indicates superior algorithm performance.

The maximum Pareto front error (MPFE) quantifies the largest distance between any vector in the approximation front and its corresponding closest vector in the true Pareto front. It assesses the dissimilarity between individual solutions in the approximation front $\hat{\pi}_i$ and the true Pareto front $\pi_k^*$. The formula of MPFE is given by:

$$MPFE = \max_k (\min_i \sum_j |f_j(\pi_k^*) - f_j(\hat{\pi}_i)|^p)^{\frac{1}{p}} \tag{10}$$

In this paper, parameter $p$ is set to 2.

The hyper area ratio (HAR) is a metric related to hypervolume (HV), which calculates the sum of the hypervolume of a hypercube formed by a given frontier and reference points. The, HAR is the ratio of the HV of the predicted problem's Pareto front to the HV of the true problem's Pareto front. The reference point is determined by the vector of the objective function values of single-objective optimal solutions. A smaller HAR signifies enhanced algorithm performance.

$$HAR = \frac{HV(\hat{PS})}{HV(PS^*)} \tag{11}$$

## A.4 THE DETAILS OF EXPERIMENTAL SETUP ON OPTIMIZATION PROBLEM

Considering the optimization problem of all testing benchmarks is multi-objective linear programming, we used the weighted-sum method to transform multi-objective problem into one single-objective linear programming, and employ the HiGHS solver in Scipy to address optimization problem. The selection of weighted-sum method to solve the multi-objective problem in this paper is due to the following reasons: for linear problems, it is provable that solutions derived from the weighted-sum method fall within the Pareto set. This can be proved by contradiction. We suppose that $\pi^w$, the optimal solution of $f_w(y, \pi)$, lies within the Pareto set of $f(y, \pi)$. Otherwise, there exists one solution $\pi^0$ in Pareto set dominates $\pi^w$, i.e, $f_t(y, \pi^0) \leq f_t(y, \pi^w), \forall t$ and $f_{t_0}(y, \pi^0) < f_{t_0}(y, \pi^w), \exists t_0$. Under this assumption, there exists one solution $\pi^0$ such that $f_w(y, \pi^0) < f_w(y, \pi^w)$. Such a result contradicts the definition of optimality for single-objective problems. Thus, in the studied optimization problems, we have proved that the optimal of $f_w(y, \pi)$ is Pareto optimal. As for

the implementation, we applied instance normalization to all objectives so as to eliminate the differences in scales between different objectives. Weights were assigned as $\frac{w}{5}$, where $w$ satisfies $\{w | \sum_i^T w_i = 5, w_i \in \mathbb{N}\}$. The set $\mathbb{N}$ denotes the set of non-negative integer.