# OpenReview forum: "Differentiation of Multi-objective Data-driven Decision Pipeline"
_ICLR.cc/2025/Conference — Submitted to ICLR 2025_

### Official Review · Reviewer_9gLK · 2024-11-03

**Soundness:** 2
**Presentation:** 2
**Contribution:** 2
**Rating:** 3
**Confidence:** 4

**Summary:**

The paper addresses the challenges of decision-focused learning (DFL) in the context of multi-objective optimization (MOP), which often involves unknown coefficients and conflicting objectives. In this setting, a new framework is proposed that enables end-to-end training of predictive models, incorporating decision loss derived from the optimization process itself. The loss functions consist of three components: landscape loss, Pareto set loss, and decision loss. Experimental results indicate that the proposed methods outperform traditional two-stage approaches and many contemporary decision-focused techniques.

**Strengths:**

This paper investigates DFL within MOP, presenting a novel problem to solve. The proposed method achieves state-of-the-art performance across multiple benchmark datasets and metrics.

**Weaknesses:**

1. The last paragraph of the paper exceeds the page limit. According to the policy, it should be desk rejected.
2. The paper is difficult to follow. The problem description lacks clarity, and the technical challenges in MOP are not well articulated, making the motivation for the proposed methods unclear. This issue is evident in all three loss components.
3. The technical contribution of the paper is not sufficiently defined. While a full page is dedicated to describing sRMMD, the derivations are largely taken from the source. The second and third loss functions are relatively straightforward, making their novelty hard to assess. This concern is underscored by ablation results indicating that the majority of performance improvement derives from the sRMMD loss.

**Questions:**

1. What is $d$ in the Pareto loss?
2. What are $r_1​$, $r_2$​, and $r$ in the experiments?

---

### Official Review · Reviewer_ueZf · 2024-11-04

**Soundness:** 2
**Presentation:** 2
**Contribution:** 2
**Rating:** 1
**Confidence:** 3

**Summary:**

This paper appears to exceed the page limit.

**Strengths:**

--

**Weaknesses:**

Your paper's main text exceeds the conference requirements, extending to page 11, which is one page over the maximum limit of 10 pages. According to the link https://iclr.cc/Conferences/2025/CallForPapers, it states "New this year, the main text must be between 6 and 10 pages (inclusive). This limit will be strictly enforced. Papers with main text on the 11th page will be desk rejected. The page limit applies to both the initial and final camera ready version." By reading your paper, we can get a general idea of ​​its structure. So, maybe consider improving lines 76 to 82 to make them more concise.

**Questions:**

--

---

### Official Review · Reviewer_xNt8 · 2024-11-05

**Soundness:** 2
**Presentation:** 2
**Contribution:** 2
**Rating:** 3
**Confidence:** 3

**Summary:**

This work designs some loss functions by considering solution space, objective space, and decision quality, named landscape loss, Pareto set loss, and decision loss, respectively for the multi-objective decision problem solving, and evaluate in two types of datasets compared to six methods.

**Strengths:**

They consider some loss functions to capture the discrepancies between predicted and true decision problems.
They explore a particular case of web advertisement allocation within the Anonymous App, aiming to optimize overall click metrics and enhance user visitation on the following day.

**Weaknesses:**

1.This work proposes empirical loss functions for multi-objective decision problems without providing theoretical guarantees, and its novelty is questionable.
2.The comparison methods used in this study are outdated and do not represent the current state-of-the-art solutions for this problem.
3.The experimental validation is limited, with few datasets and relatively basic experiments, making it difficult to substantiate the method's effectiveness.
4.The related work section only covers literature prior to 2022, lacking analysis and comparison with current research developments.

**Questions:**

1.Please provide a more comprehensive and up-to-date literature review to better contextualize the method's novelty and utility within the current research landscape.
2.Additional experimental validation on more datasets is needed to demonstrate the method's effectiveness. Furthermore, comparisons with recent state-of-the-art methods should be included.
3.The manuscript exceeds ICLR's 10-page limit for the main content.

---

### Meta-Review · Area_Chair_QYnv · 2024-12-05

**Metareview:**

This paper aims to propose a multiobjective decision-focused approach. In order to better align with the inherent properties of multi-objective optimization problems, this paper proposes a set of novel loss functions. These loss functions are designed to capture the discrepancies between predicted and true decision problems, considering solution space, objective space, and decision quality, named landscape loss, Pareto set loss, and decision loss, respectively.

The proposed method lacks the comparison with SOTA methods. Besides, this paper also exceeds the page limit of ICLR. Therefore, this paper is rejected without review.

**Additional Comments On Reviewer Discussion:**

This paper exceeds the page limit of ICLR, so the reviewers did not review this paper. This paper should be directly rejected.

---

### Decision · Program_Chairs · 2025-01-22

Reject